# Systematic evidence and gap map of research linking food security and nutrition to mental health

Thalia M. Sparling [1] ✉, Megan Deeney [1], Bryan Cheng[2], Xuerui Han[2], Chiara Lier [2], Zhuozhi Lin[3], Claudia Offner[1], Marianne V. Santoso[4], Erin Pfeiffer[5], Jillian A. Emerson[6], Florence Mariamu Amadi[7], Khadija Mitu[8], Camila Corvalan[9], Helen Verdeli[2], Ricardo Araya[10] & Suneetha Kadiyala [1]

Connections between food security and nutrition (FSN) and mental health have been analytically investigated, but conclusions are difficult to draw given the breadth of literature. Furthermore, there is little guidance for continued research. We searched three databases for analytical studies linking FSN to mental health. Out of 30,896 records, we characterized and mapped 1945 studies onto an interactive Evidence and Gap Map (EGM). In these studies, anthropometry (especially BMI) and diets were most linked to mental health (predominantly depression). There were fewer studies on infant and young child feeding, birth outcomes, and nutrient biomarkers related to anxiety, stress, and mental well-being. Two-thirds of studies hypothesized FSN measures as the exposure influencing mental health outcomes. Most studies were observational, followed by systematic reviews as the next largest category of study. One-third of studies were carried out in low- and middle-income countries. This map visualizes the extent and nature of analytical studies relating FSN to mental health and may be useful in guiding future research.

Food security and nutrition (FSN) are key components of global health and development. Internationally, healthy diets are increasingly reported to be out of reach[1] and unaffordable[2] for people of lower socioeconomic status, leading to undernutrition (e.g., wasting, underweight, micronutrient deficiency, growth faltering) in low- and middle-income countries (LMIC) and nutrition-related chronic diseases (NRCD) in both LMIC and high-income countries (HIC)[3]. Despite progress in reducing overall hunger and food insecurity (especially in Asia and Africa), one in ten people were exposed to severe levels of food insecurity in 2019, with areas or populations experiencing much higher prevalence[4]. However, in most regions, improvements in food security have slowed (including West Asia and North Africa) or reversed (including Latin America and the Caribbean) in recent years[5]. Linear growth measures are slow to reduce in line with global development goals[6], and one in every three people are overweight or have obesity[7].

Mental health has also been identified as a major cause of disability[8], although efforts to address global mental health burdens in low-resource settings is not commensurate with the magnitude of that burden[9]. Depressive disorders alone are thought to be the single-most contributor to health loss globally (7.5% of all Years Lived with Disability—YLD)[10]. Anxiety and stress, which along with depression are the

[1]London School of Hygiene and Tropical Medicine, Keppel Street, London WC1E 7HT, UK. [2]Global Mental Health Lab, Teachers College, Columbia University, New York, NY, USA. [3]Department of Mental Health, Johns Hopkins Bloomberg School of Public Health, Maryland, MD, USA. [4]Department of Anthropology, Northwestern University, Evanston, IL, USA. [5]Independent Consultant, Winston-, Salem, NC, USA. [6]Vitamin Angels, Santa Barbara, CA, USA. [7]Food for the Hungry, Phoenix, AZ, USA. [8]Department of Anthropology, University of Chittagong, Chittagong, Bangladesh. [9]Institute of Nutrition and Food Technology, University of Chile, Santiago, Chile. [10]Health Service and Population Research Department, Institute of Psychiatry, Psychology and Neuroscience, King's College London, London, UK. ✉e-mail: thalia.sparling@lshtm.ac.uk

common mental health disorders, are also leading causes of disability[11]. Despite improvements in measuring global mental health burdens, estimating the true burden remains a serious challenge. Transcultural identification and underreporting (especially due to stigma and differing social constructs) hinder the ability to make accurate global estimates[12].

Each of these fields has evolved in the last several decades. Both have shifted from clinical and continuum of care frameworks to include influential factors of wider environments and contexts, leading to an understanding of complex and systems-driven aetiologies[12]. Furthermore, the connections between FSN and mental health have been increasingly investigated. Food insecurity has been shown to lead to poor mental health in many contexts[13,14]. There is mixed or poor quality evidence linking distinct nutrients to mental health[15–17]. Dietary patterns and diet quality have been shown to be related to depression and in some instances anxiety, although heterogeneity of different measures and indices hampers the inferences we can make[18–21]. The association between BMI and mental health has perhaps been the longest-standing topic of inquiry, although this literature is dominated by research carried out in HIC settings[22,23]. Poor mental health of parents, particularly mothers, has been associated with low dietary diversity, lack of micronutrients, anthropometric outcomes, and other illness and care measures of their children in several settings[23–27], but not in others[28,29]. Mental health, for instance depression, has also been shown as a factor influencing nutritional risk and malnutrition (the nutritional aspects of frailty) in older adults[30,31]. Each of these investigations are further nuanced by their varying populations of interest and settings.

Systematic reviews on these topics are often (by nature) narrow in scope–usually in specific populations, using a particular subset of FSN and mental health indicators. Primary studies are often post-hoc or ad-hoc analyses derived from observational studies where FSN and mental health relationships are not primary outcomes. This limits the breadth and quality of the available evidence. Taking stock of the literature across interrelated aspects of FSN and mental health overall will allow for better identification and use of the strongest available evidence and more systematic efforts to research these intersections. It will also offer the possibility of creating an empirical framework that can guide hypothesis testing and causal identification going forward.

We aimed to systematically identify and map analytical studies associating FSN with mental health resulting in an interactive Evidence and Gap Map (EGM) that can offer both broad and granular views of this diverse body of literature. Our objectives were to describe the nature and range of evidence on (a) a wide range of constructs of food security and nutrition (food security, nutritional risk, diets, nutrient intakes, nutrient biomarkers, infant and young child feeding [IYCF], birth outcomes, and anthropometry), (b) linked to all types of common mental health problems (depression, anxiety, stress, and mental wellbeing), (c) across most healthy populations, settings, and study designs.

## Results

### Search and screening results
The study selection process is shown in the PRISMA Flowchart (Fig. 1). A search of three databases retrieved 40,192 results total, 30,896 of which remained after removing duplicates and were screened on title and abstract. Of these, 3771 were included for full-text review. Most articles excluded at this stage were excluded on FSN measurement, in populations with underlying health conditions, were not analytical, or were non-systematic reviews, theses, comments, or abstracts. Finally, 1945 studies met the inclusion criteria and were mapped, as shown in the HTML map linked to this article. The cells in the EGM are segmented into population groups: children (green), pregnant women and mothers (blue), adults (yellow), and mid- to later-life populations (red). Summary statistics presented here forth are not additive to the

total number of reports included, as many studies included multiple measures, populations, and settings. A simplified heat map of FSN and mental health studies is shown in Fig. 2.

### Food security and nutrition measures
Proportionally, the FSN measures in studies by group were comprised of: anthropometry (40%), diets (24%), nutrient intakes (14%), birth outcomes (13%), food scarcity (12%), nutrient biomarkers (10%), and IYCF indicators (6%).

Overall, BMI was the main indicator in 703, or 36% of all mapped studies, and was measured in almost 90% of studies including anthropometry. Studies measuring dietary patterns and quality (16%) and specific food groups (12%) were both prevalent. Of the studies measuring nutrient intake – via foods or supplements (14%), most were about macronutrients ($n = 152/273$), of which 94/152 were about polyunsaturated fatty acids (PUFA). The second largest group was vitamin intake ($n = 110/273$ studies). Of 110 studies on vitamins, various B vitamins (65%), calcium (40%), and vitamin C (29%) were most common. Of all nutrient intake studies, 87 measured supplement intake. Studies on nutrition-related birth outcomes ($n = 245$) primarily measured birth weight (84%). The majority of studies on food scarcity ($n = 230$) measured food security (71%) via many different indices. The most popular was the United States Department of Agriculture (USDA) scale used in national surveys in the US or adapted to other countries such as Canada or Korea ($n = 70$ including all versions). A small number of studies measured food scarcity through famine exposure ($n = 9$), and nutritional risk was mostly assessed in older populations ($n = 70$). Of the nutrient biomarkers in studies ($n = 202$), about half were on vitamins (55%), particularly for vitamin D (66%), folate (25%), and vitamin B12 (20%). Breastfeeding (including initiation, duration, or exclusivity) was the main FSN measure for nearly all IYCF studies ($n = 114/124$). A count of studies in each category is listed in Supplementary results 1.

### Mental health measures
Depression was by far the most common mental health measure, assessed in 61% of included studies. Hybrid domains of mental health–defined as capturing more than one aspect of mental health (e.g., a combination of depression and anxiety, a clinical interview for all common mental disorders)–were assessed in 26% of studies. Stress (12%), mental well-being (12%), and anxiety (10%) linked to FSN were the least studied.

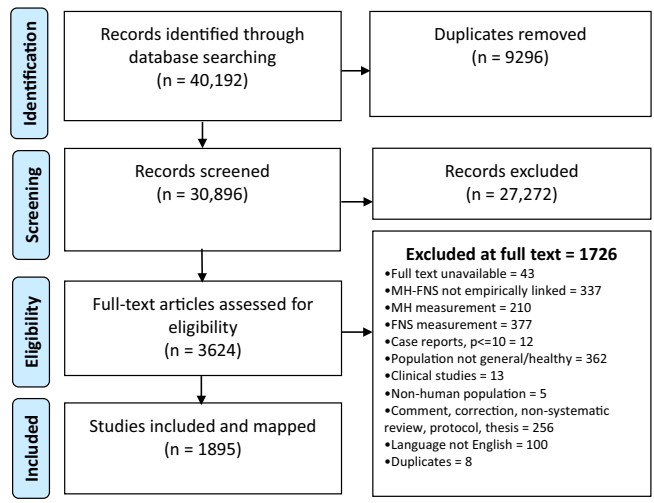

**Fig. 1 | PRISMA Flowchart.** Number of identified studies from search at each stage of screening.

| | Food scarcity | | Diets | | Nutrient intakes | | Nutrient Biomarkers | | Infant and Young Child Feeding | | Birth Outcomes | | Anthropometry | |
|---|---|---|---|---|---|---|---|---|---|---|---|---|---|---|
| | n | % | n | % | n | % | n | % | n | % | n | % | n | % |
| Depression | 126 | 6% | 278 | 14% | 205 | 11% | 170 | 9% | 95 | 5% | 129 | 7% | 455 | 23% |
| Hybrid domains (depression+) | 69 | 4% | 121 | 6% | 50 | 3% | 35 | 2% | 18 | 1% | 77 | 4% | 215 | 11% |
| Anxiety | 14 | 1% | 37 | 2% | 23 | 1% | 12 | 1% | 28 | 1% | 52 | 3% | 66 | 3% |
| Stress | 34 | 2% | 60 | 3% | 20 | 1% | 11 | 1% | 11 | 1% | 45 | 2% | 89 | 5% |
| Mental wellbeing | 30 | 2% | 69 | 4% | 22 | 1% | 14 | 1% | 4 | 0% | 14 | 1% | 110 | 6% |
| | Food security and famine, nutritional risk | | Food groups, dietary patterns and quality | | Vitamins, minerals, macronutrients, polyphenols/ antioxidants, supplements | | Vitamins, minerals, macronutrients, polyphenols/ antioxidants | | Breastfeeding, child diets and complementary feeding | | Birth weight, length, SGA/IUGR, head circumference | | BMI, body composition, body ratios, circumference, MUAC, relative height, relative weight, underweight | |
| SGA/IUGR: Small-for-gestational age, Intrauterine growth restriciton; MUAC: Mid-upper arm circumference | | | | | | | | | | | | | | |

0 — Number of studies — 300+

**Fig. 2 | Heat map of included studies.** Rows are measures of mental health, columns are measures of food security and nutrition.

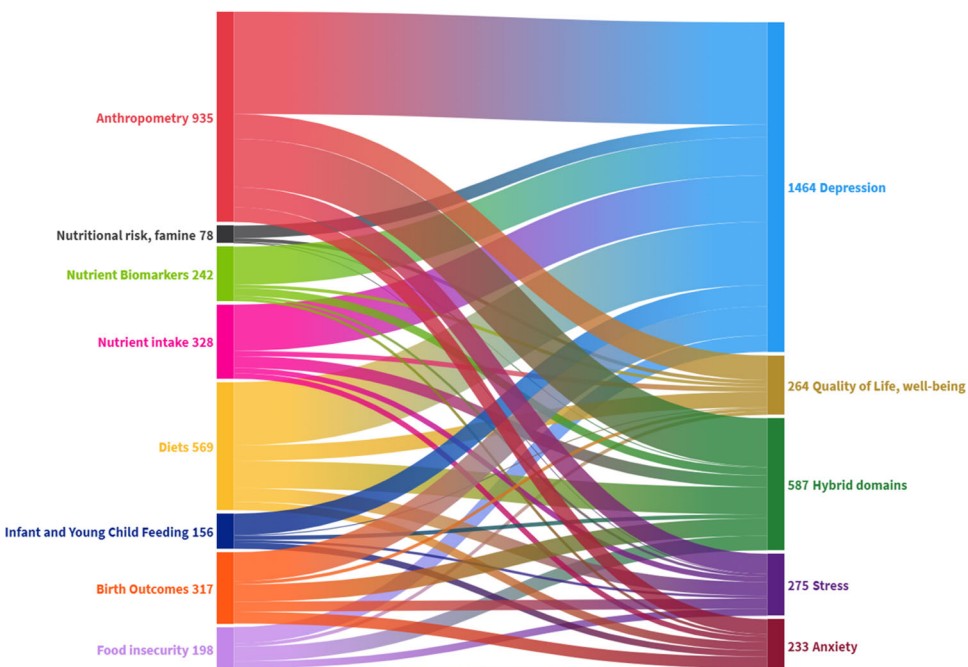

**Fig. 3 | Sankey diagram of the number of studies linking food security and nutrition (FSN) to mental health (MH).** Categories of FSN measures on the left are linked to corresponding groups of MH measures listed on the right, with the width of the bands indicating the proportional number of studies connecting the groups.

Most studies (82%) used screening questionnaires to ascertain mental health status. Mental well-being and stress have no clinical diagnosis, so almost all of these were based on established indicators via questionnaires. For depression screening, the Center for Epidemiological Studies-Depression scale (CES-D) was the most common tool (n = 332), followed by the Edinburgh Postpartum Depression Scale (EPDS) (n = 183), the Geriatric Depression Scale (n = 105) and the Patient Health Questionnaire (PHQ) (n = 104). For hybrid domains, the Global Health Questionnaire (GHQ) was the most used screening tool (n = 76), as well as the Child Behavior Checklist (CBC) for measuring mental health in children (n = 41), the Hopkins Symptom Checklist (HSCL) (n = 36) and the Depression and Anxiety and Stress Scale (DASS) (n = 33). The State-Trait Anxiety Inventory (STAI) was by far the most common screening tool for anxiety (n = 64), and the Perceived Stress Scale (PSS) and Kessler Stress Inventory (KSI) were the most common stress measures used (n = 84 and n = 46, respectively). For mental wellbeing (n = 229), 83 used the Short Form-36 questions, (also known as the Rand questionnaire). Many of these tools have been translated, adapted, and validated for cross-cultural use in LMIC contexts, and some tools have been developed specifically for these settings rather than adapted.

Clinical and diagnostic interviews were carried out in 9% of all studies, almost all of which (96%) were on depression or a general psychological or psychiatric interview which is used to diagnose multiple common mental health problems (hybrid domains). Some studies used a self-reported diagnosis, prescription medication as a proxy for diagnosis or medical records (8% of all studies). Only 14 studies investigated mental health using qualitative or mixed methods. There were 89 reviews or meta-analyses on depression, 58 on hybrid domains, 14 on anxiety, eight on stress, and three on mental wellbeing.

### Relationships between FSN and mental health

The number of studies in each FSN and MH category and the proportion investigating linkages between them are presented in the Sankey diagram in Fig. 3. The largest groups of BMI studies within anthropometry (90%) and overall (36%) were those examining BMI with: depression (n = 401, 21%), hybrid mental health measures (n = 192, 10%) and mental wellbeing (n = 109, 6%). The second largest intersection was diets (food groups, patterns, quality) with: depression (n = 278; 14%), hybrid mental health measures (n = 121, 6%) and mental wellbeing (n = 69, 4%).

Despite anthropometry and depression being the largest category, measures other than BMI and mental health besides depression were far less researched. Although there are some studies on child stunting, wasting, and underweight related to depression ($n = 45$ with depression, $n = 23$ with hybrid domains), studies reporting relationships with other common mental health disorders such as anxiety and stress were few ($n = 5$).

Although studies measuring nutrient intake were the third largest FSN group, 75% of these were analyzed for their relationship to depression, and an additional 18% to hybrid domains. Most of these studies linked macronutrients and vitamins to depression ($n = 117$ and $n = 77$, respectively), while few studies linked to anxiety, stress, or mental well-being ($n = 56$ altogether). Eighty-nine studies linked PUFA intake to depression or hybrid domains, and 32 studies to vitamin D intake and depression. There was almost an identical distribution for nutrient biomarkers, where proportionally almost all studies on biomarkers were linked to depression and hybrid measures. Vitamin D ($n = 66$) was the most common biomarker linked to depression.

Almost 50% of studies about birth outcomes ($n = 245$ total) were about birth weight with depression, and an additional 35% with hybrid domains. Many studies measured multiple nutrition-related birth outcomes (31%) such as birth length and head circumference, however only 28/245 of these included mental health measures other than depression. Only 10 of these studies investigated foetal growth restriction in relationship with mental well-being or stress, for example.

Food scarcity was linked to depression in many studies as well, especially in the studies examining nutritional risk in the elderly ($n = 56/70$). Food security was often studied in relationship to depression ($n = 72/163$), however as food security is also associated with worry, stress, and anxiety, other measures of mental health were relatively more common in the studies than in other groups of FSN (40% measured hybrid domains, 19% measured stress, 9% measured anxiety and 9% measured wellbeing).

Breastfeeding and depression were examined in 91 studies. There were especially few studies on any IYCF measure with anxiety ($n = 28$), stress ($n = 11$), and mental well-being ($n = 4$). Child diets and complementary feeding was linked to depression or hybrid domains in six out of eight child diet studies. For instance, only three studies compared any measure of mental health with child dietary diversity.

## Study methods

**Hypothesis testing.** We included studies that hypothesized the relationship between FSN and mental health in either direction: with FSN constructs as the 'exposure' or independent factor and mental health as the 'outcome' or dependent factor and vice versa (shown in each iteration, segmented proportionally by study design, in Fig. 4). Most studies ($n = 1291$, 66%) hypothesized FSN constructs as the exposure or equivalent, including cross-sectional studies. Almost 28% of these studies were about BMI associated with depression or hybrid domains of mental health outcomes. Another 25% were about diets related to depression or hybrid domains of mental health.

Mental health was treated as the exposure in 31% of studies ($n = 600$). Of these studies, 39% investigated mental health related to BMI as an outcome, of which 121 studied depression as an exposure, 69 studied hybrid domains of mental health, 60 studied stress, 27 studied anxiety, and 9 studied mental wellbeing. Birth outcomes were the second-largest group of mental health exposure studies, where 119/147 were about birth weight. Where IYCF was the outcome ($n = 75$), almost all were about breastfeeding ($n = 67$). There were relatively fewer studies on diets, nutrient intakes, and biomarkers than in either the EGM overall or where mental health was the outcome.

In a small number of studies ($n = 54$), investigators tested the hypothesis for relationships in both directions over time. For instance

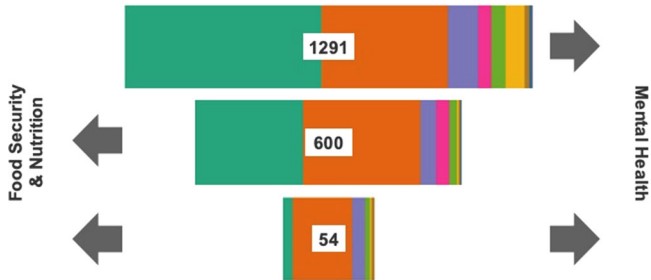

**Fig. 4 | Direction of hypothesis in included studies, segregated by study design.** The top panel is the number of studies with food security and nutrition (FSN) as the hypothesized exposure and mental health as the studied outcome. The middle panel is the number of studies with mental health as the exposure and FSN as the outcome, and the bottom panel is the number of studies where both hypotheses were investigated.

in a longitudinal cohort where dietary patterns could be isolated as an exposure among people who develop mental health problems, or alternatively within the same study population, those whose dietary patterns change over time linked to preceding mental health problems. Most of these studies investigated BMI and mental health ($n = 31/54$). These characteristics can be selected through the filter function on the interactive EGM.

**Study design.** The majority of studies were observational (83%), with 46% cross-sectional and 37% longitudinal (Supplementary results 2). An additional 3% of studies were case-control design. There were 142 systematic reviews, of which 48 offered a meta-analysis. Experimental studies were not common −only 65 Randomized Controlled Trials (RCTs) were identified, along with 20 quasi-experimental studies (12 of which used Mendelian Randomization or genetic instrumental variable methods). Only ten qualitative studies were identified, and 4 mixed methods studies, despite explicitly including qualitative eligibility and coding parameters.

Cross-sectional studies followed a similar pattern to the EGM as a whole on mental health measures, although regarding FSN there were proportionally more studies on food scarcity and BMI and fewer on birth outcomes and IYCF. There were proportionally more longitudinal studies on birth outcomes (double across all but one mental health category) and more IYCF studies, and less on nutrient intake, nutrient biomarkers, and food scarcity, although mental health measures were similar proportionally to the full EGM.

Systematic reviews and meta-analyses on diets linked to depression or hybrid domains were most common (reviews without meta-analysis = 28; reviews with meta-analysis = 9, meta-analysis without review = 4), and nutrient intakes with depression or hybrid domains were the second most common (systematic reviews = 42; 15 of these with meta-analyses). Almost all (14/15) meta-analyses on nutrient intakes were about supplements. There were 18 reviews on BMI and depression or hybrid domains (seven of these with meta-analysis), while nine others focused on child growth measures. There were 22 systematic reviews on mental health related to birth outcomes, 17 of which were about mental health of mothers and birth outcomes of their offspring. Of all 69 meta-analyses, 59 of them focused on depression or hybrid domains.

Most experimental studies were RCTs of nutrient intake exposures and mental health outcomes ($n = 46/65$ experimental studies), namely depression ($n = 26$) and hybrid domains ($n = 16$). Half of experimental studies included anxiety, stress, or mental well-being. Nutrient intakes were primarily measuring supplement intake ($n = 38/47$), especially those on B vitamins, Vitamin D, Zinc, and fatty acids.

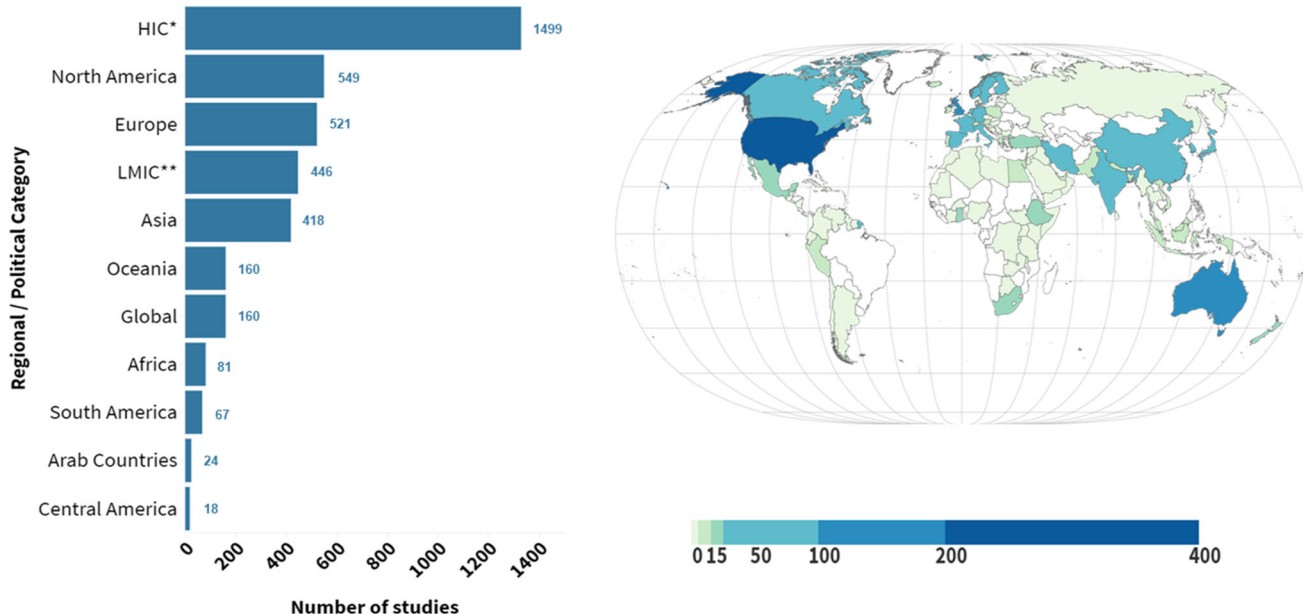

**Fig. 5 | Choropleth Map and bar plot of studies by geographic distribution.** The bar plot on the left shows the number of studies by region and political category, and the map on the right shows number of studies by country.

Sixteen RCTs exposed people to fatty acids, and 12 to Vitamin D. Several studies also exposed people to Vitamins A, C, or E and magnesium or manganese minerals. The second most common type of RCTs were those randomizing people to diets and measuring various measures of mental health (six on depression, 10 on hybrid domains, three on anxiety, seven on well-being, but none on stress). Sixteen studies intervened on: Mediterranean diet pattern ($n = 4$), low fat or low-calorie diet ($n = 4$), the DASH diet, high-protein diet, healthy diet, or fish/animal source foods ($n = 2$ each), low glycaemic diet, high protein diet and vegetarian diet ($n = 1$ each). Only three studies had mental health interventions with FSN outcomes: two on stress reduction interventions and BMI or food intake, and one on antenatal depression interventions and birthweight/child growth.

**Setting**
The geographic distribution of studies by country, defined by where the participants were located, is shown through a choropleth map in Fig. 5. The most saturation (number of studies) was in the United States, Australia, and the United Kingdom, 521 studies came from across Europe, 418 from Asia, and 81 from Africa. Central and South America were represented in fewer studies ($n = 18$ and $n = 67$ respectively). Overall, 23% ($n = 446$) were set in low- and middle-income countries (LMIC). Eight percent ($n = 160$) were 'global' studies, such as those in five or more nations across regions, or those using global datasets, such as the Gallup poll or World Bank data.

Heat maps segregated into HIC and LMIC evidence is provided in Supplementary results 3. Overall, there were proportionally more studies on nutrient intakes in HIC (15% vs. 9% of FSN measures), and proportionally more studies on food scarcity in LMIC (18% vs. 10%).

For instance, there were proportionally more studies of BMI in HIC (95% of 611 studies) compared to LMIC (72% of 172 studies). In LMIC studies, there were more studies on relative height (20% vs. 1%) and relative weight (11% vs. 2%) in children. For mental health measurement, 82% of studies using validated diagnostic tools were from HIC. Studies including measures of anxiety, stress, and mental well-being were more common in HIC than LMIC (13% vs. 7% for mental well-being).

Almost all reviews and meta-analyses were global in nature. Eighty percent of experimental studies and 90% of quasi-experimental

studies came from HIC. Populations of interest in studies from HIC were proportionally more focused on general or representative adult populations (52% vs. 42% in LMIC studies). LMIC studies shift focus to women (particularly pregnant women and mothers−35% of LMIC studies vs. 26% HIC studies), although only slightly more on children (39% of LMIC studies vs. 35% HIC studies). Studies on mid-to-later-life populations were similar in both HIC and LMIC contexts (21%).

**Populations**
The EGM linked to this paper is segmented in each cell by broad population categories. We also offer a more granular classification of populations of children, women, men, and pregnant women and mothers (available as filters). Figure 6 shows a bubble diagram proportional to the population groups of included studies. Almost half of studies in the EGM were conducted in general or representative adult populations (49%). Studies including only mid- to later-life populations (usually 60 or 65 years of age and older) made up 21% of the EGM. Of the studies that included children of any age ($n = 695$), 433 included children under 5 years, 221 included children 5 to 12 years old, and 248 focused on adolescents 13 to 18 years old. Children under 5 were not commonly assessed on their mental health status ($n = 106$ vs. 423 studies of under-five measurements of FSN) as these measures are difficult to obtain and not reliable in very young children. Pregnant, perinatal women, mothers, and fathers were studied in 28% of all studies. Far more studies in pregnant women and mothers measured mental health as the exposure than FSN (26% vs. 8%). Pregnant and postpartum women were assessed more on their mental health status (9% pregnant and 5% postpartum) than on their FSN status (3% pregnant and 1% postpartum). Studies with women-only populations (not including perinatal women or mothers) made up an additional 8% ($n = 158$). Studies focusing only on men were fewer ($n = 42$, 2%).

Some studies measured FSN in one group (e.g., children) and mental health in another (e.g., parents) (Fig. 7). Amongst these ($n = 484$), the mental health of pregnant women and parents and the FSN of their children through adolescence has been studied the most: 355 total studies, 329 on FSN of children under five years, 44 on FSN of children 5–12, and 17 on FSN of adolescents. Fathers, however, are only included in eight of these studies. Mental health of pregnant women and mothers has mostly been hypothesized as the exposure for FSN

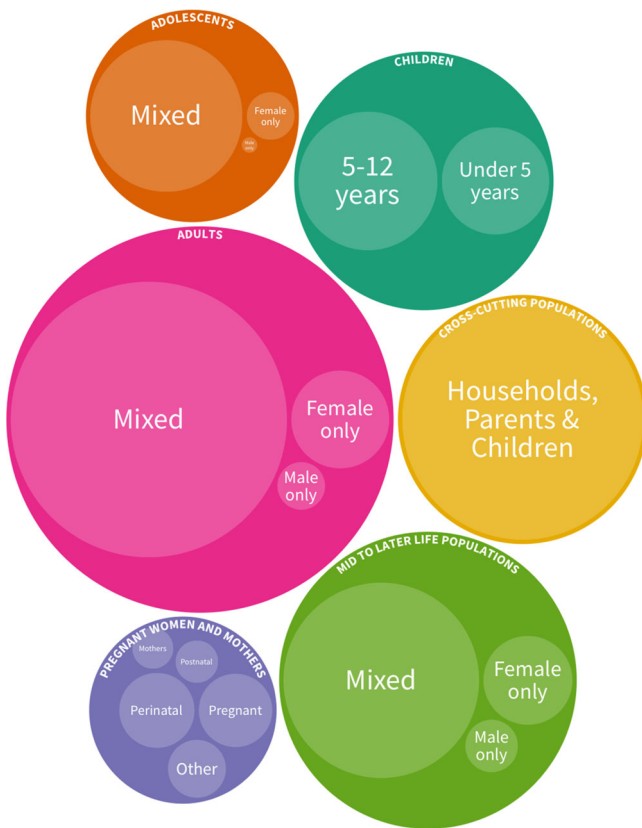

**Fig. 6 | Nested bubble diagram showing the frequency of study population groups in which FSN and mental health linkages are investigated.** Bubbles are proportional to the frequency of analyses based on each population group. Bubbles for 'Children' (n = 257), 'Adolescents' (n = 214), 'Pregnant Women and Mothers' (n = 149), 'Adults' (n = 735) and 'Mid Later Life populations' (n = 408) refer to studies in which the relationship between FSN and mental health is examined within the same study population group. The bubble for 'Cross-cutting populations' shows studies in which the FSN measure in one group is hypothesized to affect the mental health of another group or vice versa, this includes interactions between households, parents, and/or children.

outcomes in children (n = 314), though far fewer considered an association whereby FSN in children is the exposure and mental health of pregnant women and parents is the outcome (n = 54). The association between food security measured in the household with mental health in individuals was reported in 107 studies, most of which were in general adult populations (n = 51) and pregnant women and mothers (n = 38).

### Time trends
Our analysis shows clearly that the overarching body of literature linking FSN to mental health has steadily grown since 2000 (Fig. 8). As we concluded our search half-way through 2020, the number of these studies is likely to increase annually, marking a continued interest in this cross-section of fields.

## Discussion
Evidence is steadily growing about links between many of the FSN and mental health constructs measured by included studies, and the EGM makes this clear. Studies on depression and studies on BMI dominated the map overall. Anxiety, stress and mental wellbeing, and IYCF were the least represented in the literature. Given that food insecurity, inaccessibility of healthy, diverse diets, and poor clinical nutrition are all likely to exacerbate worry and stress, the dearth of studies linking FSN to dimensions of anxiety, stress, and well-being, rather than depression alone, is notable. There may be strong evidence on how food security, certain nutrients (e.g., Vitamin D), dietary patterns, and BMI are associated with depression. On the other hand, evidence seems sparse on the relationships between other nutrients (e.g., selenium, antioxidants), IYCF practices, or child growth related to mental health, or vice versa.

Regarding study design, experimental studies were mostly about nutrient intakes; very few intervened on other FSN measures or mental health interventions with FSN outcomes. Overall, experimental, quasi-experimental studies, and systematic reviews with meta-analyses were far less common than the plethora of cross-sectional and cohort studies. Only 34% of systematic reviews were accompanied by a meta-analysis. There was much less qualitative or mixed methods evidence.

Geographically, studies with paticipants from the United States, Australia, and United Kingdom dominated the evidence. Although almost a quarter of studies were carried out in LMIC, 77 of these 446

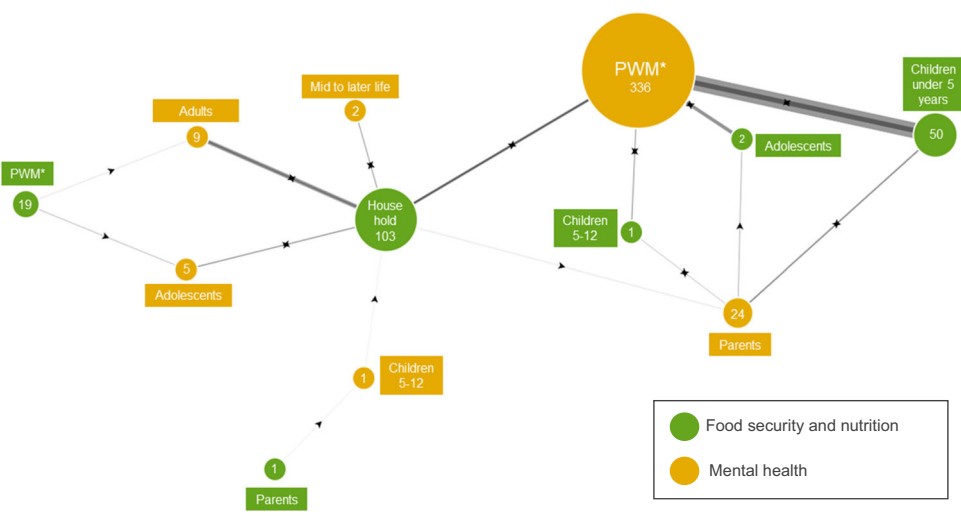

**Fig. 7 | Network diagram of studies showing the frequency of investigated relationships between FSN in one population group and mental health in another group by the hypothesized direction of the relationship.** The size of the bubbles and width of the links between them is scaled according to the number of studies and frequency of hypothesized relationships in the literature. The direction of the arrows indicates the hypothesized direction of effect according to the studies, a double arrow in opposite directions shows that both directions have been hypothesised in different studies.

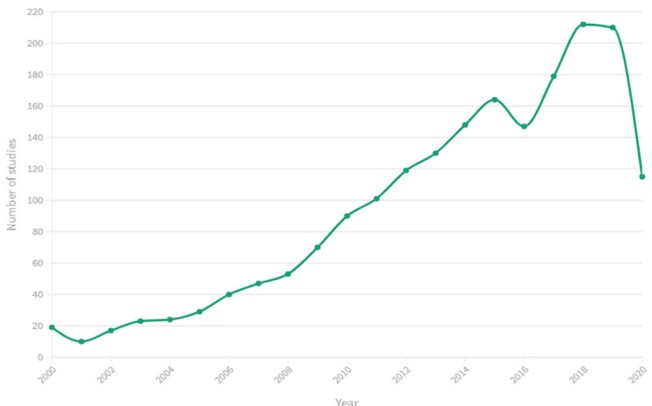

**Fig. 8 | Trends over time of analytical studies linking food security and nutrition to mental health.** The plotted line shows the increase in studies from 2000 until 2020. The search concluded half-way through 2020, which accounts for the drop off in this year.

were conducted in China and 75 in Iran, with few in Arab countries or Latin America. The studies with participants from Africa ($n = 81$) were mostly carried out in three countries (South Africa, Ghana, and Ethiopia). Three-quarters of studies carried out in South America were from Brazil. Of the LMIC countries represented in the EGM, evidence is largely based in industrialised countries, which suggests that the LMIC literature does not capture the diversity of less industrialized, poorer, or more rural countries. It is an especially important gap, given that food insecurity and undernutrition are the highest in the countries least represented by the literature base.

Most studies that measured FSN in one population group and MH in another were about mothers' mental health and their children's nutrition or growth status. Very rarely were FSN indicators in children investigated for their effect on parents' mental health. Fewer studies still focus on fathers or parents together. As studies among women in LMICs can sometimes focus on reproduction, and without sufficient attention to other aspects of womens health, we highlight the lack of studies from LMICs that examine mental health impact on women's nutritional status and vice versa.

Despite studies showing that FSN and mental health are related in many ways, there are still large gaps across the EGM of studies investigating causal mechanisms of these relationships. There were many studies showing relationships between FSN and mental health, but less with the combined design, contextual factors, and analysis to provide information most needed to design effective programs and policies. For example, there were few qualitative studies identified, even though the ethnographic lens of lived experience can provide important insights into why and how mental health is related to FSN, without relying on nosological distinctions that may be less important in certain contexts. Some of the qualitative studies raised interesting findings, for example the mental health toll from weighing trade-offs in types of food purchases (e.g., healthier options versus volume or calories)[32], how rising food prices affect not just food security and nutrition, but contribute to multi-fold mental health consequences from constraining cultural practices like funerals and other ceremonies[33], and the varied role of social support related food insecurity: in some contexts social connectedness increased shame and stigma, whereas in others it helped buffer the negative effects of food insecurity through shared resources[34].

That said, there is scope to further investigate the shared and underlying determinants of FSN and mental health. From the existing literature, these include poverty (although interestingly poverty alone does not account for these burdens[35]), lack of women's agency, other health conditions, environment, and climate change, as well as

conditions of violence, conflict, instability, and social strife[36–39]. Most of these factors have been identified through the respective bodies of literature on each, but some new work on the topic has tried to understand common determinants and mechanisms between FSN and mental health through innovative theoretical framing, study design, and more advanced statistical models[28,40]. Recent interventions that at the least measure and at the most include programmatic components of both FSN and mental health have begun to give insight into some of these mechanisms as well[41].

Through this systematic synthesis and mapping, we were able to combine various intersections of measures, populations, study types, and cross-cultural settings into an interactive resource. This is the first paper to systematize the body of evidence linking FSN to mental health. The EGM can be used in various ways by selecting and describing the nature and extent of literature on this topic.

We employed rigorous, expert-led screening and coding processes, including a search strategy designed by an information specialist using an index list of known literature. We followed state-of-the-art guidance on creating EGMs, which stop short of offering a synthesis effects observed but do include interactive filters to sort evidence according to study characteristics. Conducting a meaningful and feasible quality assessment of almost 2000 studies or pool results was beyond the scope of this EGM.

We also created parameters that limited our analysis in certain ways. We searched only papers published from 2000, did not search non-English repositories or include grey literature, and our chosen databases may not have been as likely to include qualitative reports, all which may have introduced some bias. That said, we are confident that collectively, the large number of studies identified and included serve as a basis from which to draw conclusions about trends, gaps, and characteristics of the available evidence on FSN and mental health.

The most important exclusion criteria were for studies in populations with underlying health problems, such as diabetes, cardiovascular disease, HIV, tuberculosis, or hospitalized patients, as well as niche characteristics (e.g., female endurance athletes or male textile factory workers). Although there is literature relevant for these populations, we aimed to identify evidence that minimized the confounding nature of other health conditions or characteristics. We also excluded FSN measures that were not direct measures of food security, intake, or nutrition status, such as eating behaviours, stimulant foods, or breastfeeding intentions.

In line with current trends to measure mental health globally through a symptom-based framework rather than a diagnostic criterion (which can bias and confound locally appropriate constructs of mental health)[42–44], we included mental well-being and mental health quality of life measures. We also included qualitative literature on the topic, which might not fit within the traditional depression, anxiety, and stress groupings. For instance, a systematic review of qualitative literature about depression experience globally found that DSM model and standard instruments derived from the DSM fall short of capturing the experience of depression worldwide or regionally. Specifically, half of the 15 features of depression identified in non-western populations were not captured in current diagnostic tools[42]. However, measures of mental well-being were often difficult to disentangle from general happiness, life satisfaction, or other physical health quality of life measures. Many were mixed across these domains. We thus relied on expert guidance from Teachers College Global Mental Health Lab, who assessed each measure identified across all categories for eligibility and classified them.

We propose that this EGM is a tool to navigate a diverse literature base that will be primarily driven by the interests and expertise of the user. It can identify key gaps in the literature and thus direct novel efforts in research. This might include planning new primary studies or synthesis of existing primary research. When interpreting cells with

fewer studies, it is important to carefully examine the quality of those studies and the clinical or practical relevance of research efforts to fill the gaps. Some research may be less strategic from a policy and planning perspective, for instance conducting new studies on IYCF related to anxiety and stress may have more application than new studies on minerals related to mental wellbeing, both of which appear as gaps on the EGM.

Furthermore, a cluster of studies in a cell (particularly certain study types—such as RCTs and reviews—commonly deemed further up on the hierarchy of evidence) still might prove worthy of further investigation. For instance, the most common subject of studies in the EGM is adiposity and depression, and there are several large, rigorous reviews with meta-analyses included on this topic. However, there is no pooled analysis of this relationship in low-income settings, where the observed effects may be quite different. This example highlights that the EGM as a whole can bring focus to understudied regions or populations: if used to highlight broad contextual factors, this might spur research that changes the conclusions we draw from either combining all available evidence (which may not all act in the same direction) or making assumptions based on the most prevalent literature (e.g., from high-income settings).

The overarching goal of building the EGM was to lay the groundwork for an evidence-based, empirical framework highlighting linkages that are known and hypothesized between FSN and mental health. This would entail selecting and synthesizing the strongest evidence within each cell, insofar as combining certain groups of studies is appropriate. This will serve to direct and support future inquiries into these relationships, as well as systematize our knowledge on the topic (Supplementary discussion 1, Box 1). Furthermore, a new understanding of and emphasis on these relationships can become part of advocacy, programs, strategic planning, and policy to support progress towards health goals such as the SDGs and others.

Through a systematic literature search, we comprehensively identified analytical studies investigating relationships between a broad array of FSN and mental health constructs. We mapped 1945 eligible studies onto an interactive EGM which can provide visualization of this diverse field of literature. The EGM overall allows readers to step back and take stock of the body of literature, as well as dive into specific intersections of food security, nutritional risk, diets, nutrients, nutrition-related birth outcomes, IYCF indicators, and anthropometry with depression, anxiety, stress, and mental wellbeing. The EGM also allows for narrowing of each intersection through an extensive list of filters that can be combined in various ways to select characteristics of interest.

The analysis and map highlight thematic trends (such as the proliferation of evidence linking BMI and depression) as well as gaps (stress and mental well-being related to nutrients or child diets). It also shows the nature of the literature—an increasing number of studies on the topic that are dominated by observational designs in high-income countries. Studies from Central and South America, Arab nations, and Africa are less prevalent, as well as studies using qualitative, mixed, quasi-experimental and experimental methods. Many different populations are investigated through this wide array of studies, although studies comparing associations between populations are dominated by mothers and their children.

We imagine that this analysis and EGM will serve as a basis for future inquiry, whether it be original research, evidence synthesis, and analysis, funding priorities, or the development of synergistic and integrated public health programmes and policies.

## Methods

This systematic Evidence and Gap Map, including accompanying analysis, relied on publicly accessible documents as evidence, without including personal, sensitive, or confidential information from participants, thus complying with current ethical standards.

### Search strategy

Following PRISMA guidelines, we conducted a systematic search of three published literature databases: Web of Science, CAB Global Health, and PsychInfo, searching from January 1 2000 until July 28, 2020. We chose the year 2000 as a cut-off as preliminary searches revealed diminishing returns in the eligibility and relevance of previous studies in this area. Broadly, the search was operationalized by including synonyms for mental health, stress, distress, anxiety, depression, or mood disorders, and synonyms for food security, micronutrients, diet, nutrition, or anthropometry, as well as all kinds of study designs. Results from the searches were deduplicated and loaded into EPPI Reviewer 4 and web-based software. All analysis and graphics were produced in Excel version 16 or the web-based Flourish Studio. The full search strategy, designed by an information specialist, is specified in Supplementary methods 1. The screening and coding guidelines are listed in Supplementary methods 2a–d.

### Eligibility—Inclusion

We included only papers published in peer-reviewed journals and in English, from 2000 until July 28, 2020, that presented empirical links between measures of food security and nutrition and mental health in human populations from anywhere in the world. We only included analytical research (studies associating mental health to FSN), excluding descriptive or prevalence studies. We included population-based quantitative and qualitative studies of any design. We included systematic reviews based on their eligibility criteria; to be included, at least one study in the review had to fit our overall eligibility criteria.

We included any quantitative indicator for: food scarcity (including food security, exposure to famine or hunger, and nutritional risk [usually in the elderly]); diets (specific food groups and dietary patterns or quality); nutrient intake (including vitamins, minerals, macronutrients, polyphenols/antioxidants via food intake or supplements); nutrient biomarkers (vitamins, minerals, macronutrients, and polyphenols/antioxidants measured through blood, urine, fat); Infant and Young Child Feeding (standard WHO indicators as well as breastfeeding initiation, duration or exclusivity); nutrition-related birth outcomes (e.g., birth weight, birth length, intrauterine growth restriction [IUGR] or small-for-gestational age [SGA], head circumference); and nutrition-related anthropometry (e.g., BMI, body composition, body ratios, relative weight, relative height). We used 'relative weight' as an umbrella group for wasting and weight-for-height z-scores (WHZ) and 'relative height' as a group including stunting, height-for-age z-score (HAZ), growth faltering, and other height measures of child growth. We also included studies that measured these elements of food security and nutrition through qualitative methods.

For mental health, we included studies that measured common mental disorders (CMDs) under the International Classification of Diseases version 10 (ICD-10), as well as general distress and mental well-being in order to capture transcultural and qualitative literature on the intersections of mental health and FSN. We used the following broad categories: depression; hybrid domains; anxiety; stress; and mental wellbeing (e.g., mental health-related quality of life). These could be assessed through qualitative interviews, screening questionnaires, self-report of diagnosis, prescription medication (as a proxy for diagnosis), or clinical and/or diagnostic interviews. The list of eligible screening measures was assessed and categorized by the mental health specialists at the Global Mental Health Lab.

### Eligibility—Exclusion

We did not include grey literature in our search. Studies in populations with comorbid health conditions, such as hypertension, diabetes, HIV,

or surgical patients were excluded as both the nutritional and mental health correlates of these populations is likely to be unique. We also excluded studies in populations where all participants were already identified as overweight or having obesity, low birth weight, or having mental illness. We excluded case reports ($n < 10$), theoretical or simulation-based modelling, studies in solely clinical setting, non-systematic reviews, theses, commentaries, and abstracts.

On FSN, we excluded studies on: dietary practices and attitudes without intake measures (e.g., eating family dinners, dieting); amino acids, hormones, single, specialized or stimulant foods (e.g., arginine, seaweed, walnuts only, coffee, caffeine, alcohol); proprietary or specialized supplement or food formulas; attitudes or preferences related to infant and young child care; preterm birth (as often an outcome of non-nutritional factors); and weight change, loss or trajectories. A full list of included and excluded measures with examples and justification are included in Supplementary methods 3a, b.

On mental health, we excluded mental illnesses other than CMDs (e.g., compulsive disorders, trauma-related stress disorders, phobic anxiety disorders, and developmental disorders). Measures that had no experiential component were excluded. Measures of cortisol were excluded as this hormone fluctuates for various reasons besides experience of stress (e.g., early in the morning, during birth, during exercise), as well as stressful event inventories or circumstances without ascertainment of perceived impact. General happiness or satisfaction measures were excluded as they are not direct measures of mental health, rather an indication of heightened risks or protective factors. We also excluded general health-related quality of life focusing only on physical health without mental health components separated. Lastly, we excluded studies where common mental illness could not be disentangled from other mental illness such as psychosis, bipolar disorder, substance use, eating disorders, or other mental health problems.

Some of our FSN or mental health measures (especially BMI) were included as covariables in studies for which they were not the main outcome or exposure of interest. Studies that did not report results directly linking FSN to mental health were therefore excluded.

### Screening and study selection

A team of screeners were trained and double-screened reports on title and abstract until 85% agreement rate was reached, whereafter 85% of reports were single-screened and at least 15% (sometimes more with sensitivity checking) were double-screened by a senior researcher. Patterns and disagreements were discussed and additional written guidance offered. Eligible reports based on title and abstract were reviewed in full text. We undertook a similar training process, whereby once agreement rates were reached, screeners were allowed to single screen. A third of records were double screened to ensure good sensitivity. In addition to this, several iterations of backchecking and targeted searches were re-screened throughout the process.

### Data coding and analysis

Data was classified through a mix of a priori and iterative coding strategies. Fields that were decided a priori (e.g., groups of FSN and mental health measures, countries, study designs, etc.) served to identify both trends and gaps. Iterative coding included the specific measures within FSN and mental health groups. For example, although we had pre-identified a list of common and validated measures of anxiety or depression, or food security, there were many more measures that emerged beyond initial lists. These were grouped into a code if more than one study employed the measure. We used a coding form built in EPPI Reviewer to extract data on eligible reports. Only analytical comparisons and their characteristics were considered for data extraction.

We extracted information on publication year, country (or countries) and regions, study design, hypothesized direction of association between FSN and mental health (exposure-outcome relationship) and specific categories of measures and indicators, study population characteristics and sample size, and whether the analysis was adjusted or not (with at least two covariables). For the hypothesized relationship, we coded based on the authors' stated aims and methods even for cross-sectional and qualitative studies. The 'adult' population category included any age range over 18, whereas studies with populations limited to older people (usually 60 or 65+ years old) were coded with 'mid- to later-life populations only'.

Data extraction was carried out by single coding of included studies with a full review of all data extraction forms by a second researcher and targeted sensitivity checks. Given the breadth of evidence included and the aims of an evidence and gap map, quality appraisal of individual studies was not feasible or meaningful at this stage.

>All studies that met the inclusion criteria were mapped into an EGM using standard methods[45]. The EGM framework consists of columns of categories and sub-categories of FSN constructs, and rows of mental health constructs as well as measurement categories. These rows and columns are collapsed (as the map opens) and then expanded to see all sub-categories. The cells can be segmented into four groups indicated by different colours. The bubbles scale proportionally to the number of studies in the group. The user can scroll over a cell to see a summary of studies or click on the cell to see a classified bibliography of selected studies. There is also a list of filters (codes), which can be used to select studies with specific characteristics for which data was extracted. A full coding structure is provided in Supplementary results 1.

### Reporting summary

Further information on research design is available in the Nature Research Reporting Summary linked to this article.

## Data availability

All scientific reports included in the Evidence and Gap Map were identified via Web of Science, PsychInfo, and CAB Abstracts Global Health repositories. The dataset (essentially included studies) generated during the current study are available within the HTML Evidence and Gap Map, and analysed within the manuscript and supplementary files. The full database (including initial search results and screening codes) can be accessed upon reasonable request from the corresponding author, as this is contained within EPPI Reviewer software which requires a user account.

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

## Acknowledgements

We thank the IMMANA team for their ideas, logistical and dissemination support, especially Sylvia Levy for supporting the ANH Academy Mental Health Working Group. Maria Palar, Lambert Felix, Venus Mahmoodi, Vildana Hodzic, Pema Payang, Srishti Sardana, Elliot Golden, and Justine

Wright each contributed to the screening and coding of articles and we wholeheartedly thank them for their contributions. Herbert Aimiani and Nadine Seward also contributed to the ANH Academy Working Group on Mental Health which produced this work. Funding for this study was provided by the Innovative Methods and Metrics for Agriculture, Nutrition and Health Actions (IMMANA) Programme, funded by FCDO and the Bill and Melinda Gates Foundation, which specifically funded the time of TS, MD, CO, and SK. We received in-kind support from the Global Mental Health Lab at Teachers College, Columbia University.

## Author contributions

TS conceived of the EGM and led the review, along with support from the ANH Academy Mental Health Working Group, consisting of TS, BC, MD, MS, EP, JE, FMA, KM, CC, HV, RA, and SK. TS, MD, and BC oversaw the methods and training for study identification. Screening and coding of studies was carried out by MD, TS, XH, CL, ZL, CO, and BC. TS drafted the manuscript, map, and figures, supported by CO and MD. The manuscript was reviewed by all authors, with further editing and revision support from CO, MD, BC, and SK.

## Competing interests

The authors declare no competing interests.
