## [Peer Review File · Nature Communications]

Systematic evidence and gap map of research linking food security and nutrition to mental healthREVIEWER COMMENTS

Reviewer #1 (Remarks to the Author):

Noteworthy results:

The authors' finding that this body of literature focuses almost exclusively on depression at the exclusion of other conditions (e.g. anxiety) points to a real gap in the literature and is a valuable finding of this review. It stands to reason that some of the hallmark features of food insecurity—such as worry about where food will come from—might fall more into an anxiety than a depression state, and I think this is something the authors could speculate about a bit more in the discussion and/or conclusion.

Another key insight emerging from the paper is the dearth of studies in LMICs, especially in Asia, Central/South America, and Africa, the continents with the greatest prevalence of FI. I again encourage the authors to point this out in the discussion and/or conclusion a bit more prominently. (See my critiques below for a note about a section on “gaps in the literature.”)

Comments/critiques:

First, the authors describe this as a study of “analytic research”, but unless I missed it in the paper, they don't define what they mean by this. It could be that their working definition is this statement on line 529-30: “[studies that] presented empirical links between measures of food security and nutrition and mental health in human populations.” If that's correct, I would further ask the authors to explain how they operationalized “empirical.” This might sound like a nitpicky comment, but the inclusion or exclusion of studies in a review like this one pivots on these fine-grained definitions. It might help readers better understand the scope of literature being reviewed here if the authors define terms “analytic” or “empirical” in the main body of the text.

Given my own research, the authors' apparent unquestioning acceptance of global health metrics jumped out at me as something they might at least want to acknowledge with a sentence or two. For instance, in the introduction, there's no critical discussion of the difficulty of estimating the global burden of mental disorders, which is a highly contested topic in transcultural psychiatry and psychological anthropology. Also, is it strictly correct, as the authors state in the first paragraph of the intro, that FI has increased in recent years? I know for sure that it has decreased in some places, so they might want to be careful about making a global claim like this. Somewhere around line 75 of the first page of the intro might be a good place to include a statement noting the fundamental difficulty of global surveillance of complex, affect-mediated states like mental health and even perceptions of food sufficiency or insufficiency.

In conjunction with that point, there is a fairly substantial body of ethnographic and qualitative work on lived experiences food insecurity and mental health coming out of the medical social sciences. Although some of those publications may have been included in the review, those perspectives don't make it at all into the prose of the paper, except when noted as a relative absence compared with the number of quantitative studies (e.g. lines 401-05). Again, this is arising from my own disciplinary background, but I do think that body of work provides an important and enriching counterpoint to the more quantitative work that is the focus here. While it might be a small subset of the studies reviewed in this paper, the authors' analysis would be enriched by drawing one or two key points from that body of qualitative literature to help guide future directions/suggestions, perhaps in the paragraph starting on line 407. By atomizing the elements and predictors of FI/MH, it's almost as if the paper elides the “why” or “how” questions that are central to the body of research at hand. That seems ultimately counterproductive to their aims.

As a reader of systematic reviews, I typically look for (and benefit immensely from) a section in the discussion that specifically addresses gaps in the literature and/or future directions for research. The authors hint at these topics throughout the paper, but I believe a standalone section—perhaps with its own subheading—would be helpful for readers.

Figure 3's caption could be expanded a bit to direct readers how to interpret it. Something as basic

as, 'the thickness of connecting lines represents the number of studies...' could be sufficient (perhaps similar to what appears in the caption for Figure 7).

Figure 7 didn't add much for me; if there is a need to remove a figure, I would take that one out.

Finally, is the EGM itself actually going to be provided to readers? If so, I wasn't able to access it as part of this review, so the authors should know that I did not see it. Line 120 mentions an "HTML map linked to this article," but it's not clear to me based on the materials I received what that might look like.

All of these critiques aside, thank you for this compelling work!
Dr. Lesley Jo Weaver, University of Oregon

Reviewer #2 (Remarks to the Author):

This study addresses an important public health topic, food security and mental health, which cuts across different disciplines. However, I have some reservations about the novelty of the manuscript as such, and suggest several changes:

1. The study addresses many different aspects of food security and mental health, and the search strategy could be described in greater detail.
2. While many different aspects of mental health are covered, it is surprising that ADHD, which has been found to be associated with diet and food insecurity in observational and interventional studies (Shareghfarid et al, 2020; Khoshbakht et al, 2020; Lu et al, 2019). Why has ADHD not been included in the review?
3. Food security issues may vary widely across individuals' age (in particular issues in children, pregnant women, and older adults) and should be taken into account in the review.
4. Issues related to food security are very different in high and middle-low income countries - hence stratified analyses would be relevant and this issue should be discussed in detail.

Dear Reviewers,

Thank you very much for the thoughtful feedback on our manuscript. Below you will find specific responses to each query or suggestion, and we hope that you find the overall work much improved.

Please reach out with any questions or clarifications.

Reviewer #1 (Remarks to the Author):

Noteworthy results:

- The authors' finding that this body of literature focuses almost exclusively on depression at the exclusion of other conditions (e.g. anxiety) points to a real gap in the literature and is a valuable finding of this review. It stands to reason that some of the hallmark features of food insecurity—such as worry about where food will come from—might fall more into an anxiety than a depression state, and I think this is something the authors could speculate about a bit more in the discussion and/or conclusion.
 - o Thank you for this apt comment and fair point. We have included a new sentence in the opening of the discussion to address this:
 - L 377-380: “Given that food insecurity, inaccessibility of healthy, diverse diets, and poor clinical nutrition are all likely to exacerbate worry and stress, the dearth of studies linking FSN to dimensions of anxiety, stress and wellbeing, rather than depression alone, is notable.”
- Another key insight emerging from the paper is the dearth of studies in LMICs, especially in Asia, Central/South America, and Africa, the continents with the greatest prevalence of FI. I again encourage the authors to point this out in the discussion and/or conclusion a bit more prominently. (See my critiques below for a note about a section on “gaps in the literature.”)
 - o Again, this is a fair point. We have added the following sentence to the paragraph on geography in the discussion, and as well to Box 1:
 - L400-401: “It is an especially important gap, given that food insecurity and malnutrition are the highest in the countries least represented by the literature base.”
 - Box 1: “Geography: Research on mental health and FSN relationships in low-income countries, least-developed settings, fragile and conflict-affected communities”
 -
 - o Furthermore, we do mention this in various places in the discussion, such as:
 - L489-497: “For instance, the most common subject of studies in the EGM is adiposity and depression, and there are several large, rigorous reviews with meta-analyses included on this topic. However, there is no pooled analysis of this relationship in low-income settings, where the observed effects may be quite different. This example highlights that the EGM as a whole can bring focus to understudied regions or populations: if used to highlight broad contextual factors, this might spur research that changes the conclusions we draw from either combining all available evidence (which may not all act in the same direction) or making assumptions based on the most prevalent literature (e.g. from high-income settings).”

Comments/critiques:

- First, the authors describe this as a study of “analytic research”, but unless I missed it in the paper, they don't define what they mean by this. It could be that their working definition is this statement on line 529-30: “[studies that] presented empirical links between measures of food security and nutrition and mental health in human populations.” If that's correct, I would further ask the authors to explain how they operationalized “empirical.” This might sound like a nitpicky comment, but the inclusion or exclusion of studies in a review like this one pivots

on these fine-grained definitions. It might help readers better understand the scope of literature being reviewed here if the authors define terms “analytic” or “empirical” in the main body of the text.

- Thank you for raising this. We meant analytical in the sense that any descriptive or prevalence studies without explicit cross-section of MH to FSN measures would be excluded. We have clarified this in the following way in the methods:
 - L553-555: “We only included analytical research (studies associating mental health to FSN), excluding descriptive or prevalence studies.”
- Since the methods come at the end of the paper, we also included this small note in the aims:
 - L109-110: “We aimed to systematically identify and map analytical studies associating FSN with mental health...”
- Given my own research, the authors’ apparent unquestioning acceptance of global health metrics jumped out at me as something they might at least want to acknowledge with a sentence or two. For instance, in the introduction, there’s no critical discussion of the difficulty of estimating the global burden of mental disorders, which is a highly contested topic in transcultural psychiatry and psychological anthropology.
 - Please see a response to this point below (L74-77).
- Also, is it strictly correct, as the authors state in the first paragraph of the intro, that FI has increased in recent years? I know for sure that it has decreased in some places, so they might want to be careful about making a global claim like this.
 - Thanks for pointing this out. We have amended the introductory sentences to clarify this:
 - L63-66: “Despite progress in reducing overall hunger and food insecurity (especially in Asia and Africa), one in ten people were exposed to severe levels of food insecurity in 2019, with areas or populations experiencing much higher prevalence (4). However, in most regions, improvements in food security have slowed (West Asia and North Africa) or reversed (Latin America and Caribbean) in recent years (5).”
- Somewhere around line 75 of the first page of the intro might be a good place to include a statement noting the fundamental difficulty of global surveillance of complex, affect-mediated states like mental health and even perceptions of food sufficiency or insufficiency.
 - This is a very good point and we are happy to expand on this as you suggest. Thank you very much for suggesting a placement as well!
 - L74-77: “Despite improvements in measuring global mental health, estimating the true burden remains a serious challenge. Transcultural identification and underreporting (especially due to stigma and differing social constructs), hinder our ability to make accurate global estimates (11).”
 - We do also come back to this point in the discussion as it relates to our methods, and added to the existing text:
 - L466-470: “In line with current trends to measure mental health globally through a symptom-based framework rather than a diagnostic criterion (which can bias and confound locally appropriate constructs of mental health) (42-44), we included mental wellbeing and mental health quality of life measures. We also included qualitative literature on the topic, which might not fit within the traditional depression, anxiety, and stress groupings. For instance, a systematic review of qualitative literature about depression experience globally found that DSM model and standard instruments derived from the DSM fall short of capturing the experience of depression worldwide or regionally. Specifically, half of the 15 features of depression identified in non-western populations were not captured in current diagnostic tools (42).”

- In conjunction with that point, there is a fairly substantial body of ethnographic and qualitative work on lived experiences food insecurity and mental health coming out of the medical social sciences. Although some of those publications may have been included in the review, those perspectives don't make it at all into the prose of the paper, except when noted as a relative absence compared with the number of quantitative studies (e.g. lines 401-05). Again, this is arising from my own disciplinary background, but I do think that body of work provides an important and enriching counterpoint to the more quantitative work that is the focus here. While it might be a small subset of the studies reviewed in this paper, the authors' analysis would be enriched by drawing one or two key points from that body of qualitative literature to help guide future directions/suggestions, perhaps in the paragraph starting on line 407. By atomizing the elements and predictors of FI/MH, it's almost as if the paper elides the "why" or "how" questions that are central to the body of research at hand. That seems ultimately counterproductive to their aims.
 - o This is an important issue. We have added a paragraph highlighting this issue and offering a few examples from the qualitative literature we identified:
 - L415-423: "For example, there were few qualitative studies identified, even though the ethnographic lens of lived experience can provide important insights into why and how mental health is related to FSN, without relying on nosological distinctions that may be less important in certain contexts. Some of the qualitative studies raised interesting findings, for example the mental health toll from weighing trade-offs in types of food purchases (e.g. healthier options versus volume or calories) (32), how rising food prices affect not just food security and nutrition, but contribute to multi-fold mental health consequences from constraining cultural practices like funerals and other ceremonies (33), and the varied role of social support related food insecurity: in some contexts social connectedness increased shame and stigma, whereas in others it helped buffer the negative effects of food insecurity through shared resources (34)."

- As a reader of systematic reviews, I typically look for (and benefit immensely from) a section in the discussion that specifically addresses gaps in the literature and/or future directions for research. The authors hint at these topics throughout the paper, but I believe a standalone section—perhaps with its own subheading—would be helpful for readers.
 - o Thanks for this suggestion. We feel that we have addressed both gaps and future directions, albeit interspersed thematically throughout the discussion, as you note. Therefore, we have added a summary box to bring together key opportunities we mention. Box 1 text reads:
 - Mental health dimensions: Research that focuses on anxiety, stress, and mental wellbeing, beyond depression alone
 - FSN dimensions: Research on nutrients other than Vitamin D (e.g., selenium, antioxidants), as well as infant and young child feeding, child growth and cognitive development, especially research on these measures as exposures to long-term outcomes
 - Study design: Studies using meta-analysis, experimental and quasi-experimental studies on FSN measures other than nutrients; mixed-methods and qualitative research
 - Geography: Research on mental health and FSN relationships in low-income countries, least-developed settings, fragile and conflict-affected communities
 - Populations: Research is especially needed that examines FSN as exposures to mental health outcomes for caregivers and adults; research on fathers and parents. Research is also lacking for women's health beyond their reproductive roles.
 - Shared and underlying determinants common to both mental health and FSN, especially inequity, empowerment, social support and cohesion, and instability or conflict.

- Figure 3's caption could be expanded a bit to direct readers how to interpret it. Something as basic as, 'the thickness of connecting lines represents the number of studies...' could be sufficient (perhaps similar to what appears in the caption for Figure 7).
 - o Thanks for pointing this out. We have changed the caption to read:
 - *"Categories of FSN measures on the left are linked to corresponding groups of MH measures listed on the right, with the width of the bands indicating the proportional number of studies connecting the groups."*
- Figure 7 didn't add much for me; if there is a need to remove a figure, I would take that one out.
 - o Thanks for this suggestion. We didn't have another comment about this, but we are happy to remove it if the editor agrees that the paper would be stronger without.
- Finally, is the EGM itself actually going to be provided to readers? If so, I wasn't able to access it as part of this review, so the authors should know that I did not see it. Line 120 mentions an "HTML map linked to this article," but it's not clear to me based on the materials I received what that might look like.
 - o Indeed, it was our intention for this to be provided to you. The final version is only available as an HTML file, which we requested was specially provided to reviewers, but sometimes this gets lost in the mix. There is a pre-final version of the map at: <https://www.anh-academy.org/anh-academy/working-groups/mental-health-working-group>. We will also request again to editors that the final HTML version is distributed to you. We will request again that the EGM is provided.
- All of these critiques aside, thank you for this compelling work!
Dr. Lesley Jo Weaver, University of Oregon
 - o Thank you very much for taking the time to give a very helpful and fair review of our work, and we think that your suggestions strengthen the paper.

Reviewer #2 (Remarks to the Author):

This study addresses an important public health topic, food security and mental health, which cuts across different disciplines. However, I have some reservations about the novelty of the manuscript as such, and suggest several changes:

- The study addresses many different aspects of food security and mental health, and the search strategy could be described in greater detail.
 - o Thank you for raising this point. We offer the full search strategy in the appendix of the paper. As well, we have added the following description to the methods:
 - L542-544: "Broadly, the search was operationalized by including synonyms for mental health, stress, distress, anxiety, depression or mood disorders, and synonyms for food security, micronutrients, diet, nutrition or anthropometry, as well as all kinds of study designs."
- While many different aspects of mental health are covered, it is surprising that ADHD, which has been found to be associated with diet and food insecurity in observational and interventional studies (Shareghfarid et al, 2020; Khoshbakht et al, 2020; Lu et al, 2019). Why has ADHD not been included in the review?
 - o Thanks for inquiring about this. ADHD is not considered a Common Mental Disorder (CMD). CMDs are depressive and anxiety disorders that are classified in ICD-10 as: "neurotic, stress-related and somatoform disorders" and "mood disorders". In particular, ADHD is relatively common in children, but is rarer in adults.
 - o That said, we have clarified the inclusion and exclusion criteria sections in the methods:

- L574-577: “For mental health, we included studies that measured common mental disorders (CMDs) under the International Classification of Diseases version 10 (ICD-10), as well as general distress and mental well-being in order to capture transcultural and qualitative literature on the intersections of mental health and FSN.”
 - L603-604: “we excluded mental illnesses other than CMDs (e.g. compulsive disorders, trauma-related stress disorders, phobic anxiety disorders, and developmental disorders).”
- Food security issues may vary widely across individuals' age (in particularly issues in children, pregnant women, and older adults) and should be taken into account in the review.
 - Thanks for this important point, and we couldn't agree more. Although we explicitly requested the editor to make the interactive EGM available to the reviewers, we now understand that the actual EGM was not distributed to you as a reviewer – we are sorry for this. We will request again that this be distributed to you separately from other materials. A pre-final EGM (slightly different from the final version) can be found here: <https://www.anh-academy.org/anh-academy/working-groups/mental-health-working-group>

The EGM itself is segregated into population groups, and thus we extensively represent and classify the literature from these different age groups. Additionally, we provide the following analysis of the literature by age group of population studied:

- L336-364: “The EGM linked to this paper is segmented in each cell by broad population categories. We also offer a more granular classification of populations of children, women, men and pregnant women and mothers (available as filters). Figure 6 shows a bubble diagram proportional to the population groups of included studies. Almost half of studies in the EGM were conducted in general or representative adult populations (49%). Studies including only mid- to later-life populations (usually 60 or 65 years of age and older) made up 21% of the EGM. Of the studies that included children of any age (n=695), 433 included children under 5 years, 221 included children 5 to 12 years old, and 248 focused on adolescents 13 to 18 years old. Children under 5 were not commonly assessed on their mental health status (n=106 vs. 423 studies of under-five measurements of FSN) as these measures are difficult to obtain and not reliable in very young children. Pregnant, perinatal women, mothers and fathers were studied in 28% of all studies. Far more studies in pregnant women and mothers measured mental health as the exposure than FSN (26% vs. 8%). Pregnant and postpartum women were assessed more on their mental health status (9% pregnant and 5% postpartum) than on their FSN status (3% pregnant and 1% postpartum). Studies with female-only populations (not including perinatal women or mothers) made up an additional 8% (n=158). Studies focusing only on males were fewer (n=42, 2%).

Some studies measured FSN in one group (e.g., children) and mental health in another (e.g., parents) (Figure 7). Amongst these (n=484), the mental health of pregnant women and parents and the FSN of their children through adolescence has been studied the most: 355 total studies, 329 on FSN of children under 5, 44 on FSN of children 5-12, and 17 on FSN of adolescents. Fathers, however, are only included in 8 of these studies. Mental health of pregnant women and mothers has mostly been hypothesized as the exposure for FSN outcomes in children (n=314), though far fewer considered an association whereby FSN in children is the exposure and mental health of pregnant women and parents is the outcome (n=54). The association between food security measured in the household with mental health in individuals

was reported in 107 studies, most of which were in general adult populations (n=51) and pregnant women and mothers (n=38).”

- Issues related to food security are very different in high and middle-low income countries - hence stratified analyses would be relevant and this issue should be discussed in detail.
 - This is a fair point. We discussed this issue in the following paragraph:
 - L394-400: “Although almost a quarter of studies came from LMIC, 77 of these 446 came from China and 75 from Iran, with few from Arab countries or Latin America. The studies from Africa (n=81) mostly came from only three countries (South Africa, Ghana and Ethiopia). Three quarters of studies from South America were from Brazil. Of the LMIC countries represented in the EGM, evidence is largely based in industrialised countries, which suggests that the LMIC literature doesn’t capture the diversity of less industrialized, poorer, or more rural countries.
 - We also added to the paragraph to strengthen the point that the least developed countries, with the highest burdens of food insecurity and malnutrition, are the least represented in the literature, even among LMICs:
 - L400-401: “It is an especially important gap, given that food insecurity and undernutrition are the highest in the countries least represented by the literature base.”
 - Furthermore, we added the following sentence to the beginning of the discussion:
 - L378-380: “Given that food insecurity, inaccessibility of healthy, diverse diets, and poor clinical nutrition are all likely to exacerbate worry and stress, the dearth of studies linking FSN to dimensions of anxiety, stress and wellbeing, rather than depression alone, is notable.
 - Just to note, this EGM does not synthesize observed effects from included studies, and therefore stratified quantitative analysis is beyond the scope of this paper. That said, we are currently undertaking meta-analyses of various intersections of the literature, focusing specifically on LMIC contexts, and hope that these contributions will add to the results presented in this manuscript.

REVIEWERS' COMMENTS

Reviewer #1 (Remarks to the Author):

Excellent revisions! You have addressed my concerns thoroughly.

Reviewer #2 (Remarks to the Author):

The authors should be congratulated for thoroughly addressing all my comments.